# Equine Tenocyte Seeding on Gelatin Hydrogels Improves Elongated Morphology

**DOI:** 10.3390/polym13050747

**Published:** 2021-02-28

**Authors:** Marguerite Meeremans, Lana Van Damme, Ward De Spiegelaere, Sandra Van Vlierberghe, Catharina De Schauwer

**Affiliations:** 1Comparative Physiology, Faculty of Veterinary Medicine, Ghent University, Salisburylaan 133, B-9820 Merelbeke, Belgium; Catharina.DeSchauwer@ugent.be; 2Polymer Chemistry and Biomaterials Group, Centre of Macromolecular Chemistry, Faculty of Sciences, Ghent University, Krijgslaan 281 S4-Bis, B-9000 Ghent, Belgium; Lana.VanDamme@UGent.be (L.V.D.); Sandra.VanVlierberghe@Ugent.be (S.V.V.); 3Department of Morphology, Faculty of Veterinary Medicine, Ghent University, Salisburylaan 133, B-9820 Merelbeke, Belgium; Ward.Despiegelaere@UGent.be

**Keywords:** tenocytes, hydrogels, gelatin, cell proliferation, viability, morphology, gel fraction, swelling ratio, storage modulus

## Abstract

(1) Background: Tendinopathy is a common injury in both human and equine athletes. Representative in vitro models are mandatory to facilitate translation of fundamental research into successful clinical treatments. Natural biomaterials like gelatin provide favorable cell binding characteristics and are easily modifiable. In this study, methacrylated gelatin (gel-MA) and norbornene-functionalized gelatin (gel-NB), crosslinked with 1,4-dithiotreitol (DTT) or thiolated gelatin (gel-SH) were compared. (2) Methods: The physicochemical properties (^1^H-NMR spectroscopy, gel fraction, swelling ratio, and storage modulus) and equine tenocyte characteristics (proliferation, viability, and morphology) of four different hydrogels (gel-MA, gel-NB85/DTT, gel-NB55/DTT, and gel-NB85/SH75) were evaluated. Cellular functionality was analyzed using fluorescence microscopy (viability assay and focal adhesion staining). (3) Results: The thiol-ene based hydrogels showed a significantly lower gel fraction/storage modulus and a higher swelling ratio compared to gel-MA. Significantly less tenocytes were observed on gel-MA discs at 14 days compared to gel-NB85/DTT, gel-NB55/DTT and gel-NB85/SH75. At 7 and 14 days, the characteristic elongated morphology of tenocytes was significantly more pronounced on gel-NB85/DTT and gel-NB55/DTT in contrast to TCP and gel-MA. (4) Conclusions: Thiol-ene crosslinked gelatins exploiting DTT as a crosslinker are the preferred biomaterials to support the culture of tenocytes. Follow-up experiments will evaluate these biomaterials in more complex models.

## 1. Introduction

Overuse tendon injuries are a major cause of musculoskeletal morbidity in both human and equine athletes [1]. Approximately 30–50% of all sport lesions in both professional and recreational athletes are tendon injuries with increased age being an additional risk factor [2,3]. Injuries to the tendons situated at the palmar/plantar side of the equine distal limb are the most common orthopedic injuries in competition horses subjected to high-intensity exercise [4]. The horse is also one of the most well-accepted, scientifically supported animal models of human exercise-induced Achilles tendon injury [5]. The poor success with conventional therapy supports the need to search for novel treatments to restore the functionality and regenerate a tissue as close to the original tendon as possible, instead of inferior scar tissue [2,4].

In vitro models serve as important biological tools to study cell behavior under controlled conditions, bypassing the confounding factors associated with in vivo clinical trials [6,7]. A wide diversity of in vitro tendon models are used nowadays to study tendon mechanobiology, tissue replacement processes, cell-based treatments, and drug screening applications [8,9]. Unfortunately, no generally accepted in vitro model or standard tenogenic differentiation assay exists currently. Various biomaterials, bioreactors and production technologies are randomly combined at this moment [10]. By establishing a physiologically representative in vitro tendinopathy model, the use of experimental animals and the number of in vivo experiments can be drastically reduced.

The tendon extracellular matrix (ECM) predominantly consists of collagen I [2,11] and therefore, this natural biomaterial is mostly used for tendon applications [12,13,14]. Gelatin, being denatured collagen, contains the same peptide sequences (e.g., arginine-glycine-aspartic acid sequence) as collagen, critical for cell surface receptor recognition, and can be used as an alternative to collagen [15,16]. When compared to collagen I, other advantages of gelatin include lower immunogenicity, higher water solubility, lower cost, and wide availability, which is important when considering large-scale in vitro studies [8,17]. To provide more material strength, the gelatin backbone can be modified by introducing different moieties at the level of different amino acid side-chain functionalities [18]. For years, methacrylated gelatin (gel-MA) has been used as the gold standard [8,19,20]. Although not yet reported in the context of tendon applications, norbornene-functionalized gelatin (gel-NB) has already been proven suitable and even superior to gel-MA, in adipose tissue engineering, for example [21,22]. Furthermore, the polymerization of gel-NB combined with a thiolated crosslinker, occurs through a thiol-ene step growth polymerization instead of a chain growth polymerization (as is the case for gel-MA). Step growth polymerization is characterized by a higher reactivity, a more homogeneous end product and a more cell-friendly approach towards cell encapsulation, because of reduced radical formation [15,17,23]. It is therefore hypothesized that gel-NB combined with a thiolated crosslinker (i.e., 1,4-dithiotreitol, DTT versus thiolated gelatin, gel-SH) could be a valuable biomaterial to support the culture of tenocytes.

In order to identify a more physiologically representative hydrogel for tenocytes, which enables the expression of the characteristic tenogenic morphology, four different gelatin hydrogels were compared to tissue culture plastic (TCP) in this study (Table 1). The physicochemical characteristics, including the chemical structure via ^1^H-NMR spectroscopy, gel fraction, swelling ratio and mechanical properties of three gel-NB-based compositions were evaluated and benchmarked against gel-MA. Furthermore, tenocyte morphology and functionality were evaluated.

## 2. Materials and Methods

### 2.1. Materials

Gelatin type B, isolated from bovine skin was kindly supplied by Rousselot (Ghent, Belgium). N-(3-dimethylaminopropyl)-N′-ethylcarbodiimide (EDC), N-acetyl-homocysteine thiolactone, ethylene diamine tetra-acetic acid (EDTA), acetone, methacrylic anhydride, 5-norbornene-2-carboxylic acid, collagenase type Ia, antibiotic-antimycotic solution, trypsin-EDTA, calcein-acetoxymethyl (Ca-AM), propidium iodide (PI), the focal adhesion staining kit, paraformaldehyde, and secondary goat anti-mouse antibody were purchased from Sigma-Aldrich (Diegem, Belgium). High glucose Dulbecco’s Modified Eagle Medium (DMEM), L-glutamine, 4-well plates, Tween-20, Triton-X, and bovine serum albumin (BSA) were bought from Fisher Scientific (Merelbeke, Belgium). Dimethyl sulfoxide (DMSO) and N-hydroxysuccinimide (NHS) were obtained from Acros (Geel, Belgium). The Spectrapor dialysis membranes MWCO 12–14 kDa were purchased from Polylab (Antwerp, Belgium). Lithium (2,4,6-trimethylbenzoyl) phenylphosphinate (LAP) was synthesized according to a protocol described earlier [24].

### 2.2. Disc Development

#### 2.2.1. Development of Gel-MA

Gel-MA was obtained following a previously reported protocol by Van den Bulcke et al. [25]. Briefly, 100 g gelatin type B was dissolved in 1 L of a 0.1 M phosphate buffer (pH 7.8) at 40 °C (Figure 1). Afterwards, 2.5 equivalents of methacrylic anhydride were added dropwise and the mixture was allowed to react for 1 h under continuous mechanical stirring. Next, the reaction mixture was diluted with 1 L double distilled water (ddH_2_O), followed by dialysis (Spectrapor 12–14 kDa cutoff) against distilled water (dH_2_0) for 24 h at 40 °C, changing the water at least five times. The pH of the solution was adjusted to ~7.4. The obtained gel-MA was then frozen and lyophilized (Christ freeze-dryer alpha I-5; −80 °C; 0.37 mbar). The degree of substitution (DS) was determined using ^1^H-NMR spectroscopy (Bruker Avance WH 500 MHz).

#### 2.2.2. Development of Gel-NB

A protocol described earlier by Van Hoorick et al. was used to obtain gel-NB [22]. Briefly, 2.5 equivalents of 5-norbornene-2-carboxylic acid (relative to the primary amines of gelatin) were converted into an activated succinimidyl ester using carbodiimide coupling chemistry. To this end, 2 equivalents of EDC followed by 3 equivalents of NHS were added to dry DMSO under inert atmosphere at room temperature. Following 25 h of reaction, the mixture was heated to 50 °C. In parallel, gelatin type B was dissolved in dry DMSO at 50 °C (Figure 1). The reaction mixture was added to the dissolved gelatin and left to react overnight at 50 °C under Argon atmosphere. Next, the mixture was precipitated in a tenfold excess of acetone. The precipitate was isolated and washed with acetone, before dissolving the residue in ddH_2_O. Afterwards, the mixture was dialyzed (Spectrapor MWCO 12–14 kDa cutoff) for 24 h at 40 °C. The pH of the mixture was adjusted to ~7.4, before freezing and lyophilization of the product. The DS was assessed using ^1^H-NMR spectroscopy. In order to obtain gel-NB with a lower DS, the same protocol was applied using 1.2 equivalents of 5-norbornene-2-carboxylic acid, 0.75 equivalents EDC and 1.5 equivalents NHS.

#### 2.2.3. Development of Gel-SH

Gel-SH was developed by dissolving 20 g of gelatin type B in 200 mL carbonate buffer (pH 10) at 40 °C under inert argon atmosphere, as described earlier (Figure 1) [26]. When the gelatin was completely dissolved, 1.5 mM of EDTA was added in order to chelate any metals which could catalyze the oxidation of the sulfhydryls into disulfide bonds. Afterwards, 5 equivalents (relative to the primary amines of gelatin) of N-acetyl homocysteine thiolactone were added and the mixture was left to react for 3 h under inert atmosphere at 40 °C. Next, the obtained gel-SH was purified using dialysis (SpectraPor 12–14 kDa cutoff) for 24 h under inert atmosphere at 40 °C. The water was changed 5 times. Following dialysis, the gel-SH was immediately frozen in liquid nitrogen followed by lyophilization. The DS was determined using an ortho-phthalic dialdehyde amine determination assay.

#### 2.2.4. Hydrogel Film Casting

Prior to determining the physicochemical properties of the hydrogel discs and seeding cells, the starting materials were processed into films. To this end, a 10 *w*/*v*% aqueous solution of the gelatin derivatives, containing 2 mol% LAP as photoinitiator, was made at 40 °C. The thiolated crosslinkers, either DTT or gel-SH, were added in an equimolar quantity (1:1 thiol-ene ratio). When complete dissolution was obtained, the mixture was poured between two glass plates covered with release foil and separated by a 1 mm silicone spacer. Crosslinked hydrogels were obtained by putting the plates at +4 °C for 30 min, followed by exposure to UV-A light (365 nm, 9 mW/cm²) for 30 min.

### 2.3. Physico-Chemical Characterization 

#### 2.3.1. Gel Fraction Assessment

To determine the gel fraction, the crosslinked hydrogel discs (d = 8 mm) were freeze-dried determining the dry mass (m_d1_). Next, the dried hydrogel discs were incubated in ddH_2_O at 37 °C for 24 h, followed by freeze-drying to determine the second dry mass (m_d2_). The gel fraction was determined using the following Equation: (1)$$\mathrm{Gel}\;\mathrm{fraction}\;\left(\%\right)=\;\frac{m_{d2}}{m_{d1}}*100$$

#### 2.3.2. Mass Swelling Ratio Determination

To determine the swelling ratio, the hydrogel discs (d = 8 mm) were incubated immediately after crosslinking for 24 h in ddH_2_O at 37 °C to obtain equilibrium swelling. Next, the hydrated mass of the samples was measured (m_h_) and the samples were lyophilized to determine their dry mass (m_d_). The mass swelling ratio was determined using the following formula: (2)$$\mathrm{Mass}\;\mathrm{swelling}\;\mathrm{ratio}=\;\frac{m_{h}}{m_{d}}$$

#### 2.3.3. Mechanical Properties’ Assessment

To obtain insight into the hydrogels’ mechanical properties, a rheometer (Physica MCR-301; Anton Paar, Sint-Martens-Latem, Belgium) was used. Punched out discs (14 mm in diameter) of equilibrium swollen hydrogel films (1 mm in height, incubated for 24 h in ddH_2_O at 37 °C) were placed between the two parallel plates. Differences in storage moduli were assessed using a strain of 0.1% with an oscillation frequency ranging from 0.01 Hz to 10 Hz at 37 °C, whilst applying a normal force of 1 N. All measurements were performed in triplicate.

### 2.4. Cell Isolation, Culture and Seeding

Briefly, equine superficial digital flexor tendons were aseptically collected in the slaughterhouse and transported within 2 h to the lab. Representative tendon samples of 2−3 g were obtained and washed in PBS containing 0.5% gentamycin. After three washing steps with PBS, the tendon samples were minced into small pieces, and subsequently digested overnight in high glucose DMEM supplemented with 0.075% collagenase type Ia at 38 °C in a humidified atmosphere containing 5% CO_2_. After blocking collagenase activity with fetal bovine serum (FBS) containing medium, the suspension was filtered on a 70 µm cell strainer to remove undigested tissue. After centrifugation (400 g, 10 min, RT), two more washing steps with PBS were performed. Isolated cells were incubated at 38 °C in a humidified atmosphere containing 5% CO_2_ in tenocyte medium consisting of high glucose DMEM, 10% FBS, 1% antibiotic-antimycotic solution, and 2mM L-glutamine [27]. After 72h, the medium was removed and the cells were washed with PBS. Thereafter, the medium was replaced twice weekly. When they reached 80−90% confluency, the tenocytes were passaged using 0.25% Trypsin-EDTA. Cell viability was determined by trypan blue exclusion using the improved Neubauer hemocytometer [28]. The tenocytes of passage three (P3) were plated at a density of 40,000 cells/cm^2^ in tenocyte medium in either 4-well plates of TCP or gelatin discs. Gelatin discs of 14 mm in diameter were punched out and sterilized by submerging them in ethanol 70% for 24 h (refreshed after 12 h) followed by UV−C irradiation (30 min) before cell seeding. Tenocyte medium was added onto the discs following 1 h of incubation and was carefully refreshed twice weekly. All experiments were performed in triplicate.

### 2.5. Viability Assay

The dye solution, consisting of 2 µL Ca-AM and 2 µL PI per 1 mL PBS, was used to assess cell viability at days 7 and 14 after seeding, using an inverted fluorescence microscope (Axio Observer 7, ZEISS, Jena, Germany). Four random images per hydrogel per replicate at 488/509 nm (Ca-AM, live cells) and 584/608 nm (PI, dead cells) were evaluated. The area occupied by the cells was measured per image with an ImageJ macro (adapted from [29]) and cell viability was determined with the following formula:(3)$$\mathrm{Area}\;\mathrm{live}\;\mathrm{cells}\;\left(\%\right)=\frac{\mathrm{Area}\;\mathrm{live}\;\mathrm{cells}}{\mathrm{Area}\;\mathrm{live}\;\mathrm{cells}\;+\mathrm{Area}\;\mathrm{dead}\;\mathrm{cells}}\times 100$$

### 2.6. Cell Morphology 

A focal adhesion staining kit was used at days 7 and 14 to (1) confirm cellular alignment by visualizing the actin filaments and the focal contacts between cells, (2) identify the nuclei for counting and (3) determine their shape. Briefly, cell-seeded hydrogel discs were fixed with 4% paraformaldehyde (in PBS) for 20 min, washed twice with washing buffer (PBS + 0.05% Tween-20) and permeabilized using 0.1% *w*/*v* Triton-X (in PBS) for 4 min. After double washing, tenocytes were blocked with 1% BSA (in PBS) for 30 min, subsequently incubated with the primary anti-vinculin antibody (1:400, in blocking solution) for 1 h. After three washes of each 5 min, the secondary goat anti-mouse antibody (FITC-conjugated) (1:200, in PBS) and the TRITC-conjugated phalloidin (1:200, in PBS) were incubated simultaneously for 1 h. Counterstaining of the nuclei with DAPI (1:1000, in PBS) for 4 min was performed after three washes (5 min each). After three final washing steps (5 min each), the cells were covered with PBS for visualization. All incubations were performed at room temperature. 

Several regions of interest per condition per replicate were evaluated, images were taken at 495/519 nm, 548/561 nm and 353/465 nm, for FITC, TRITC and DAPI using an inverted fluorescence microscope (Axio Observer 7, ZEISS, Germany), respectively. An ImageJ macro (adapted from [29]) was used to quantify cell numbers per image using the DAPI channel. Additionally, cell shape was qualitatively evaluated on the actin channel, combined with quantitative determination of nucleus’ circularity and aspect ratio (AR), being major indications for an elongated cell shape (Figure 2) [30,31,32]. ImageJ analysis was performed four times and the mean of each parameter was used for statistical analysis.
(4)$$\mathrm{Circularity}=\;4\pi \times \frac{\left[\mathrm{Area}\right]}{{\left[\mathrm{Perimeter}\right]}^{2}}\;\mathrm{Aspect}\;\mathrm{Ratio}=\frac{\left[Major\;Axis\right]}{\left[Minor\;Axis\right]}$$

### 2.7. Statistical Analysis

Data are shown as mean (M) ± standard deviation (SD). Statistical analysis was performed with GraphPad Prism, (GraphPad Software; San Diego, CA, USA) and in R Studio (Version 1.3.1093, RStudio, PBC, Boston, MA, USA). For physicochemical properties one-way ANOVA tests were executed. Post hoc Tukey’s test displayed pairwise differences between the conditions. Cell characteristics, i.e., cell numbers, viability and cell morphology, were assessed with non-parametric Kruskal−Wallis tests, combined with Dunn’s multiple comparison test. Reported *p*-values were corrected for the multiple comparisons by the Bonferroni correction method. 

## 3. Results

### 3.1. Materials

#### 3.1.1. Gelatin Modification

Gelatin type B was used as starting material onto which photo-crosslinkable functionalities were introduced along the backbone (Figure 1). The frequently reported gel-MA was used as a benchmark throughout the present work. ^1^H-NMR spectroscopy was performed to determine the DS of the developed gel-MA and gel-NB (Figure 3). To this end, either the characteristic peaks of the methacrylamide moieties (5.6 and 5.8 ppm) or of the norbornene functionalities (6.33 and 6.00 ppm (endo derivative); 6.28 and 6.28 ppm (exo derivative)) were compared to the methyl signals of the side chains of the inert Val, Leu, Ile peak at 1.01 ppm. Gel-MA was successfully developed with a DS of 97%. Furthermore, gel-NB with both a high DS of 85% and a low DS of 55% were obtained, which was in agreement with previously reported literature [21,22]. The DS of gel-SH was determined using an ortho-phthalic dialdehyde assay, yielding a DS of 75%. These results are in line with earlier reports, obtaining gel-SH with a DS of 72% [21].

#### 3.1.2. Physicochemical Properties

An overview of the assessed physicochemical properties of the obtained films is shown in Figure 4. To gain insight into the amount of potentially harmful precursors leaching out, which can be harmful to the cells, the gel fraction of the hydrogel films was determined. As anticipated, films with a higher DS of gel-NB resulted in a higher gel fraction due to its denser crosslinked network. In addition, it was observed that the applied thiolated crosslinker did not influence the gel fraction, as both the gel-NB85/DTT and gel-NB85/SH75 resulted in similar gel fractions (87.5 ± 2.4% vs. 89.6 ± 1.8%). However, the gel fraction of the thiol-ene based films was significantly lower compared to gel-MA (98.8 ± 1.3%), *p* < 0.0001.

The mass swelling ratio gives an indication of the ability of the hydrogel to mimic the aqueous cellular environment of the ECM. A significantly lower mass swelling ratio was observed for gel-MA (13.7 ± 0.5) compared to the thiol-ene based films, *p* < 0.0001. The thiol-ene films were characterized by at least a 1.6 times higher swelling ratio, varying between 22.3 and 25.2. Furthermore, a higher swelling ratio was obtained for gel-NB with a lower DS compared to a higher DS, *p* = 0.0246. 

Considering the mechanical properties of the hydrogel films, a significantly higher storage modulus (up to 1.9 times higher) was observed for gel-MA (31.3 ± 3.6 kPa) compared to the thiol-ene films (varying between 16.3–17.7 kPa), *p* = 0.0007, *p* = 0002, and *p* = 0.0014, respectively. No significant differences were observed between the thiolated crosslinkers used. Furthermore, as anticipated, there was an inverse correlation between the obtained swelling ratio of the films and their mechanical properties.

### 3.2. Cell Characteristics

#### 3.2.1. Cell Proliferation

Mechanical properties, structure and composition of the hydrogel are all features which impact cellular function [13]. To assess cell proliferation on the different hydrogel compositions, the number of DAPI positive cells per image was determined (Figure 5A). At day 7, significantly more cells were observed on gel-NB55/DTT (415.54 ± 213.1) when compared to gel-NB85/DTT (153.44 ± 215.13), *p* = 0.0052. After 14 days in culture, however, the number of cells on gel-NB85/DTT (340.42 ± 84.19) was comparable with those on gel-NB55/DTT (383.08 ± 27.71). Cell numbers measured on gel-MA (187.92 ± 91.55) were significantly lower compared to thiol-ene based hydrogels (gel-NB85/SH75: 326.92 ± 80.63), *p* < 0.0001, *p* = 0.0048, and *p* = 0.0203, respectively. It can therefore be concluded that gel-MA is certainly not optimal to support tenocyte cultures. Gel-NB55/DTT also significantly differed from TCP (241.57 ± 45.87), *p* = 0.0024, which indicates a superior biocompatibility of gel-NB55/DTT, especially when considering TCP is specifically designed for cell culture.

#### 3.2.2. Viability Assay

As both living and dead cells are stained with DAPI, an additional fluorescent staining with Ca-AM/ PI was performed to assess the viability of the equine tenocytes. Viability on all hydrogels exceeded 95%, indicating excellent biocompatibility (ISO 10993-5 (2009)). It must be mentioned however that the viability for gel-NB55/DTT (97.65% ± 2.24) and gel-NB85/SH75 (97.16% ± 1.83) was significantly lower at 14 days when compared to gel-MA (99.47% ± 0.52), *p* = 0.0156, and *p* = 0.0011, respectively (Figure 5B).

#### 3.2.3. Cell Morphology

To confirm the tenogenic phenotype, morphological characterization is usually performed as no unique tendon marker has been identified yet [11]. Tenocytes should be spindle-shaped, elongated and organized in a parallel fashion [33]. Qualitative evaluation of the fluorescent images however showed a rounder cell morphology on gel-MA discs and an intermediate morphology, i.e., between round and spindle-shaped, when cultured on TCP and gel-NB85/SH75 (Figure 6). 

To confirm these findings, the nuclear shape was analyzed, as it is representative for cell shape [30,31]. If the ratio of the major nuclear axis to the minor nuclear axis, the so-called AR, is bigger than 1, this is indicative for a more elongated nuclear shape (Figure 2). A significantly higher AR was measured for gel-NB55/DTT (1.71 ± 0.11) at day 7 compared to the other materials evaluated (TCP: 1.41 ± 0.10; gel-MA: 1.44 ± 0.07; gel-NB85/DTT: 1.40 ± 0.20), *p* = 0.0004, *p* = 0.0052, and *p* = 0.0009, respectively (Figure 7A). At day 14, a significantly lower AR was measured for gel-MA (1.40 ± 0.07) when compared to other hydrogels (gel-NB85/DTT: 1.90 ± 0.12; gel-NB55/DTT: 1.87 ± 0.16; gel-NB85/SH75: 1.77 ± 0.17), *p* < 0.0001, *p* < 0.0001, and *p* = 0.0031, respectively. Circularity, indicated by a value close to 1, represents an aberrant cell shape when discussing tenocytes (Figure 2). After 7 days, cells cultured on TCP (0.77 ± 0.02) displayed a significantly rounder nuclear shape than cells on gel-MA (0.70 ± 0.05), gel-NB85/DTT (0.65 ± 0.11) and gel-NB55/DTT (0.70 ± 0.04), *p* = 0.0154, *p* = 0.0018, and *p* = 0.0008, respectively (Figure 7B). On the other hand, circularity measured at day 14 was significantly lower for all thiol-ene based films (gel-NB85/DTT: 0.65 ± 0.02; gel-NB55/DTT: 0.64 ± 0.02; gel-NB85/SH75: 0.69 ± 0.04) compared to gel-MA (0.76 ± 0.03), *p* < 0.0001, *p* < 0.0001, and *p* = 0.0165, respectively. In addition, cells cultured on gel-NB85/DTT and gel-NB55/DTT displayed a significantly more elongated cell shape when compared to TCP (0.73 ± 0.03), *p* = 0.0125 and *p* = 0.0047. In conclusion, qualitative assessment, AR and circularity data showed a favorable morphology of tenocytes when cultured on thiol-ene based films, especially for gel-NB85 and gel-NB55 combined with the bifunctional thiolated crosslinker DTT, while tenocytes cultured on TCP and gel-MA showed a less desirable morphology.

## 4. Discussion

As it is generally accepted that standard cell culture materials lack biomimetic capacities, current research is focusing on the evaluation of different biomaterials, especially when considering specific applications [13]. Parameters which are evaluated include correct biochemical composition and structure, biocompatibility towards appropriate cell populations, and appropriate mechanical strength and elasticity [12,34,35].

Indeed, physicochemical characteristics, mechanical properties, and cell characteristics are intrinsically linked. A significantly lower mass swelling ratio was observed for gel-MA discs, which resulted in inferior cell characteristics, as could be observed in Figure 5A and Figure 7 (i.e., low cell proliferation, lower AR and higher circularity). The effect of the swelling ratio has been demonstrated before on chondrogenic differentiation. Mesenchymal stem cells were encapsulated in different hydrogel compositions showing a superior chondrogenic differentiation for the hydrogel compositions with a higher swelling ratio [36]. The higher mass swelling ratio of the thiol-ene discs indeed showed superior cell proliferation (day 14) and morphology (higher AR and lower circularity), indicating that the aqueous ECM environment was well mimicked using both biomaterials. The lower swelling properties of Gel-MA can be attributed to the hydrophobic oligo-methacrylamide chains present in the crosslinked gel-MA network [37]. The higher swelling ratio obtained for gel-NB55/DTT can be explained by the fact that a lower DS results in a less densely crosslinked network [38]. Moreover, since norbornene is hydrophobic, a higher DS results in less swelling. The results are in agreement with previous literature reports, describing a lower mass swelling ratio of gel-MA hydrogels compared to thiol-ene films [17,21,39].

Both in vitro and in vivo, tenocytes respond to mechanical stimuli via cell−cell and cell−matrix interactions, and mechano-transduction pathways [7,40]. When tenocytes are exposed to appropriate mechanical stimulation, they arrange in a parallel fashion [41]. In the study of Rowlands et al., the effect of mechanical stiffness was evaluated by exposing mesenchymal stem cells to different acrylamide substrates, with increasing stiffness, of which the surface was modified using various proteins, i.e., collagen I, collagen IV, fibronectin, and laminin. Osteogenic differentiation was significantly more pronounced on stiffer materials (80 kPa), regardless of the protein surface used, whereas myogenic differentiation was affected by the protein coating and was observed on both stiff (80 kPa) and soft materials (25 kPa) [42]. As tenocytes are specialized fibroblasts, just as myoblasts, a similar response of tenocytes on gelatin might be expected. Another research group reported the highest expression of myogenic markers by mesenchymal stem cells when cells were cultured on 11kPa hydrogels, while 34 kPa gels are preferred for osteogenic differentiation [43]. Thiol-ene based hydrogels with lower storage moduli (16.3–17.7 kPa) also resulted in favorable cell characteristics in this study. Furthermore, a higher stiffness for chain-growth hydrogels (such as gel-MA) has been described in literature due to the presence of nondegradable kinetic chains of an uncontrolled length [21,37,39].

Although gel-MA hydrogels were characterized by a higher storage modulus (31.3 kPa), usually desirable for tendon tissue engineering [44], this resulted in both an inferior biocompatibility and cell morphology. It could therefore be hypothesized that the heterogeneous nature of the chain growth polymerization resulted in different mechanical cues throughout the construct which were less preferred by the tenocytes. This can be substantiated by the cells’ morphology and alignment seen in Figure 6 as the tenocytes seemed more clustered compared to the other compositions. It can be concluded that the more homogeneous step growth polymerization could offer additional beneficial results irrespective of the lower storage moduli.

In general, cell proliferation curves are sigmoidal. Immediately after seeding cells, a lag phase is observed for approximately 48 h, depending on the cell type, during which cells adapt to the new culture conditions [45]. Subsequently, cells enter an exponential growth phase, a so-called log phase. For human tenocytes, a lag phase of approximately 4 days was reported on TCP [46]. The lower cell numbers observed for gel-NB85/DTT at day 7 might be explained by an extended lag phase. However, an initial cytotoxic effect of DTT could also explain the results observed. The latter is in agreement with the literature, reporting on a potential cytotoxic effect of DTT especially for encapsulated cells [39,47]. These results can be substantiated by a gel fraction <90% and the lower cell density present at day 7, as shown in Figure 6. Nevertheless, due to the favorable material properties of gel-NB85/DTT (e.g., high swelling ratio), cells are able to proliferate and obtain a similar cell density during the log phase of the other constructs, indicating that any undesirable effects only occurred in the initial phase.

The results of the viability assay support the characteristic biocompatibility of natural biomaterials, e.g., gelatin [16]. The decreased viability observed (Figure 5B) comparing day 7 to day 14 when using gel-NB55/DTT can be explained by contact inhibition, a well-known process in which cells stop proliferating when confluency is reached [48]. As already mentioned, tenocytes cultured on gel-NB55/DTT proliferated faster, and, as such, reached confluency faster when compared to the other biomaterials evaluated in this study. Furthermore, the high proliferation rate of tenocytes cultured on gel-NB55/DTT, creating a higher cell density, also resulted in a favorable cell morphology [49].

Tenocyte dedifferentiation, indicating that tenocytes lose their typical morphology and eventually their functionality [46], is frequently reported for TCP cell cultures. The current experiment strengthened this statement by the detrimental nuclear shape on TCP (low AR and high circularity). This is possibly explained by the unphysiologically stiff characteristics of TCP [13]. In contrast, the elongated nuclear shape (high AR and low circularity) on thiol-ene based hydrogels can be explained not only by its favorable physicochemical properties, but furthermore by the tripeptide arginine-glycine-aspartic acid sequence present in gelatin, which was facilitating cell adherence and therefore cell spreading by connecting with integrins in the cell membrane [16,49].

In this study, there were two limitations which would be interesting to evaluate in future research. First, an advanced technique such as qPCR or immunostaining might be used to further evaluate cell functionality. Second, the effect of cell encapsulation should be verified. Cell encapsulation is more physiologically relevant than seeding the cells on the biomaterial and as such, more appropriate for assessing both the advantages and the disadvantages of the individual biomaterials in relation to tenocyte culture [50]. The current study focused on the evaluation of different chemically crosslinked gelatin derivatives (i.e., realized via step growth versus chain growth polymerization) and their potential to support tenocyte adhesion, viability and proliferation. Assessing functionality and encapsulating tenocytes in future experiments will improve our knowledge in order to identify the preferred gelatin derivative which can be combined with tenocytes in a 3D-environment. From this study, it is clear that gel-MA is not eligible as an appropriate biomaterial to support the culture of equine tenocytes.

## 5. Conclusions

In the present study, the physicochemical characteristics, mechanical properties, and cell characteristics of four gelatin hydrogels were evaluated. The thiol-ene based hydrogels showed a significantly lower gel fraction/storage modulus and a higher swelling ratio compared to gel-MA. Although gel-MA is frequently used as the gold standard for tissue engineering purposes, detrimental tenocyte characteristics support the search for alternative biomaterials. Thiol-ene crosslinked gelatins exploiting DTT as a crosslinker (gel-NB85/DTT and gel-NB55/DTT) emerged as the preferred biomaterials to culture tenocytes when considering both their physicochemical and the cell characteristics. The current research improves our knowledge on the interaction between natural biomaterials and tenocytes, which is essential to establish a representative tendon model.

## Figures and Tables

**Figure 1 polymers-13-00747-f001:**
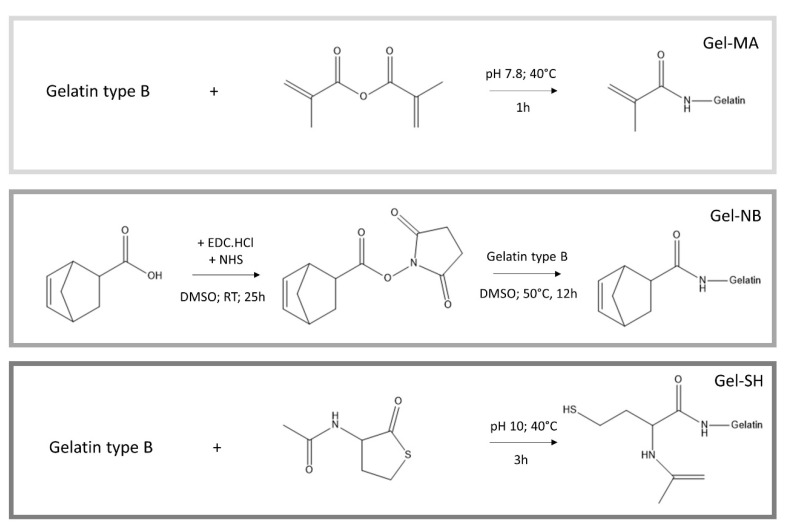
Reaction scheme depicting the development of gel-MA, gel-NB and gel-SH precursors. The crosslinked hydrogel films are obtained through chain growth polymerization for gel-MA and via step growth polymerization when gel-NB is combined with either DTT or gel-SH. DTT: 1,4-dithiotreitol; EDC: N-(3-dimethylaminopropyl)-N’-ethylcarbodiimide hydrochloride; gel-MA: methacrylated gelatin; gel-NB: norbornene-functionalized gelatin; gel-SH: thiolated gelatin; NHS: N-hydroxysuccinimide.

**Figure 2 polymers-13-00747-f002:**
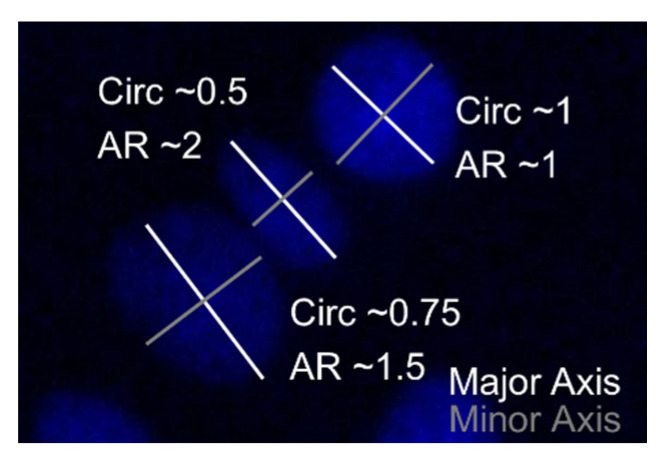
Visualization of nuclear shape on DAPI stained nuclei. If the ratio of the major nuclear axis to the minor nuclear axis, the aspect ratio (AR), is bigger than 1, a more elongated shape is seen. Circularity (Circ.) is measured between 0 and 1, the value being close to 1 represents a rounder cell shape.

**Figure 3 polymers-13-00747-f003:**
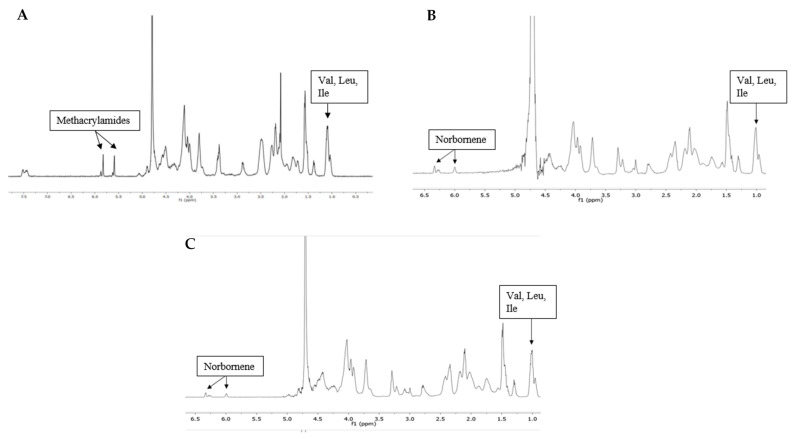
To confirm the degree of substitution, ^1^H-NMR spectra were assessed of (**A**) Gel-MA DS 97%, (**B**) Gel-NB DS 85%, and (**C**) Gel-NB DS 55%. Gel-MA: methacrylated gelatin; gel-NB: norbornene-functionalized gelatin.

**Figure 4 polymers-13-00747-f004:**
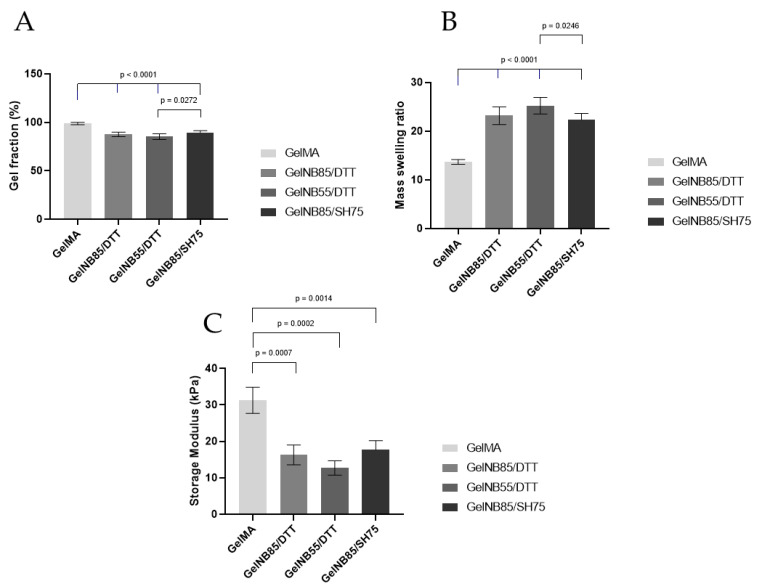
Physicochemical properties of the obtained hydrogel films: (**A**) gel fraction, (**B**) mass swelling ratio, and (**C**) mechanical properties of 10 *w*/*v*% films containing 2 mol% LAP photoinitiator. Significant differences are shown with adjusted p-values and error bars representing standard deviation. DTT: 1,4-dithiotreitol; gel-MA: methacrylated gelatin; gel-NB: norbornene-functionalized gelatin; gel-SH: thiolated gelatin; LAP: lithium(2,4,6-trimethylbenzoyl) phenylphosphinate.

**Figure 5 polymers-13-00747-f005:**
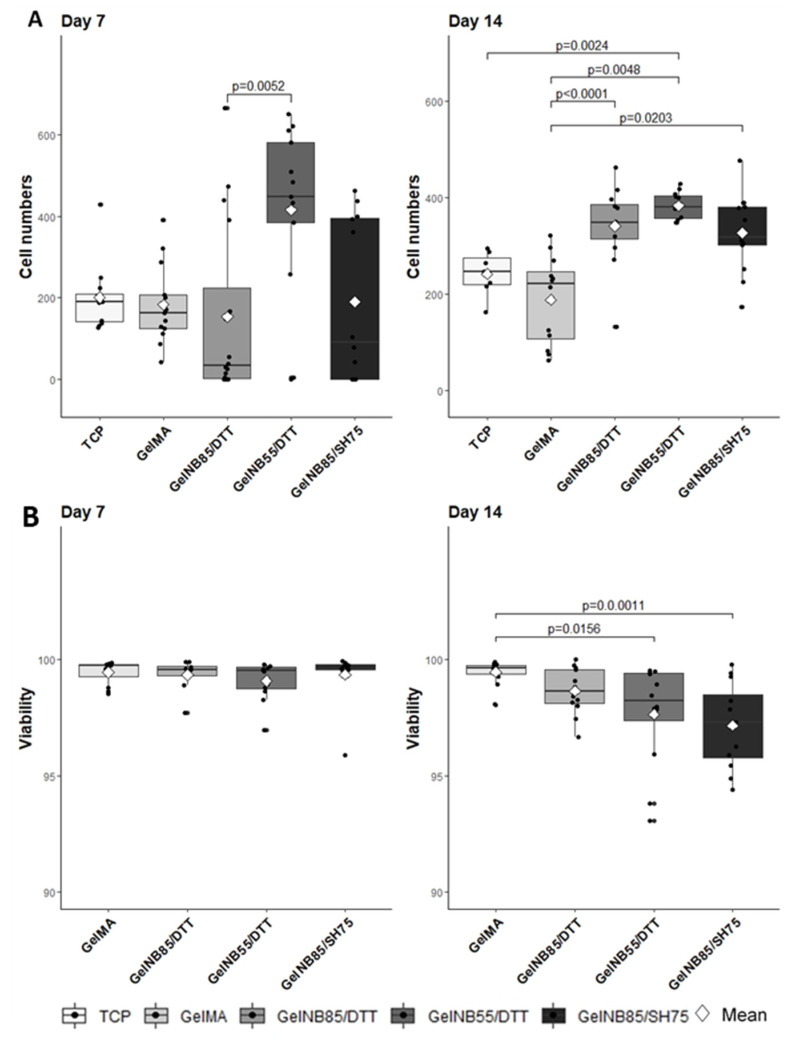
Tenocyte characteristics as determined on thiol-ene based hydrogels and compared to both TCP and gel-MA: (**A**) cell proliferation: cell numbers per image are shown, (**B**) viability assay: the percentage of the area occupied by living cells is shown in relation to the total cell area cultured per image. Significant differences are shown with adjusted p-values. After 14 days in culture the number of cells on all thiol-ene based hydrogels were comparable and significantly higher than on gel-MA. It can therefore be concluded that gel-MA is not optimal to support tenocyte cultures. Gel-NB55/DTT also significantly differed from TCP which indicates superior biocompatibility of gel-NB55/DTT. Viability on all hydrogels exceeded 95%, indicating excellent biocompatibility (>70%, ISO 10993-5 (2009)). The decreased viability on gel-NB55/DTT can be explained by contact inhibition. DTT: 1,4-dithiotreitol; gel-MA: methacrylated gelatin; gel-NB: norbornene-functionalized gelatin; gel-SH: thiolated gelatin; TCP: tissue culture plastic.

**Figure 6 polymers-13-00747-f006:**
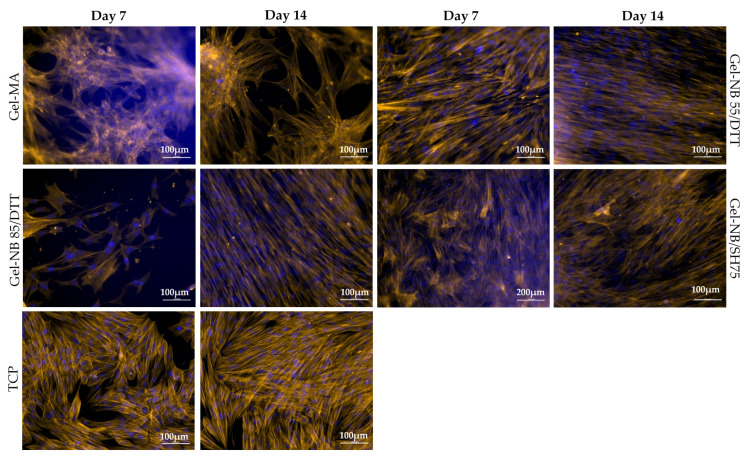
Representative images of the qualitative evaluation of tenocyte morphology and cell alignment based on fluorescent staining of the actin filaments (orange) with DAPI nuclear counterstaining (blue). Experiment was performed in triplicate. Tenocytes should be spindle-shaped, elongated and organized in a parallel fashion (as seen on gel-NB55/DTT days 7 and 14, and gel-NB85/DTT day 14). A rounder cell morphology on gel-MA discs is seen (both timepoints) and an intermediate morphology (between round and spindle-shaped) on TCP and gel-NB85/SH75 (both timepoints). DTT: 1,4-dithiotreitol; gel-MA: methacrylated gelatin; gel-NB: norbornene-functionalized gelatin; gel-SH: thiolated gelatin; TCP: tissue culture plastic.

**Figure 7 polymers-13-00747-f007:**
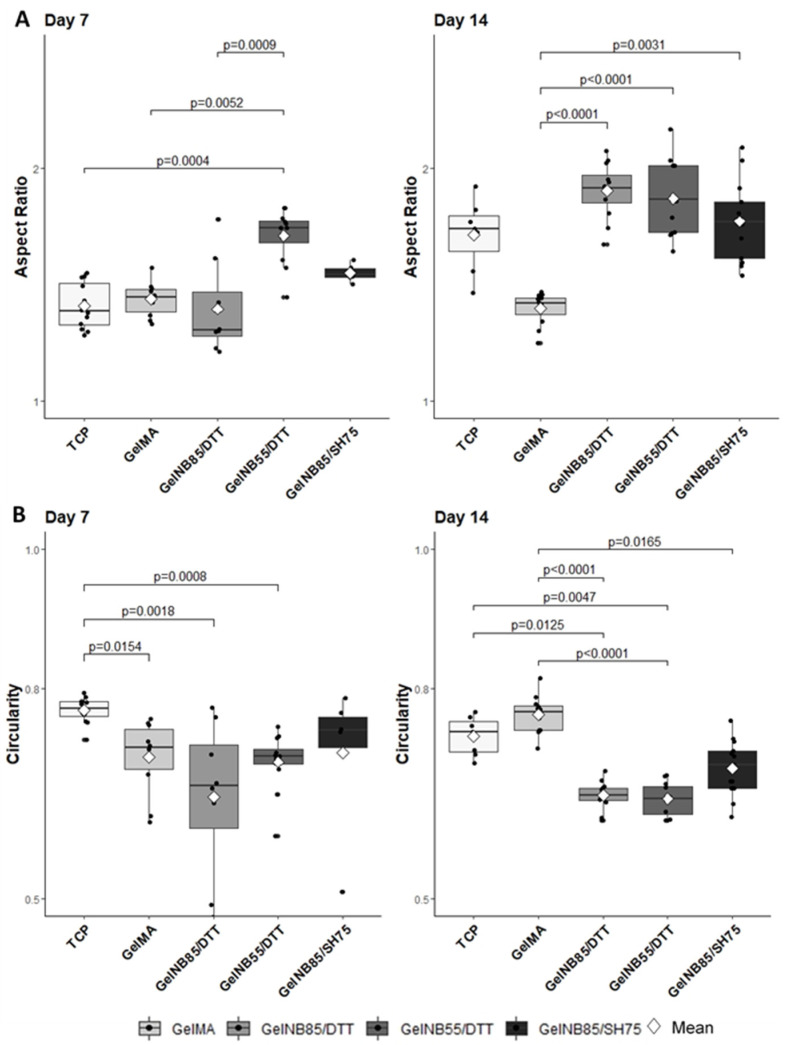
Tenocyte characteristics as determined on thiol-ene based hydrogels and compared to both TCP and gel-MA: (**A**) nuclear aspect ratio (AR): ratio of the major nuclear axis to the minor nuclear axis. An elongated cell shape is represented by a higher aspect ratio, and (**B**) circularity: a perfect circle has a value of 1. Significant differences are shown with adjusted *p*-values. Favorable tenocyte characteristics are high AR and low circularity, as expressed on all thiol-ene based hydrogels, most clearly pronounced on day 14. DTT: 1,4-dithiotreitol; gel-MA: methacrylated gelatin; gel-NB: norbornene-functionalized gelatin; gel-SH: thiolated gelatin; TCP: tissue culture plastic.

**Table 1 polymers-13-00747-t001:** Experimental set-up: four different gelatin biomaterials were evaluated as hydrogels for tenocyte culture. Tissue culture plastic was included as control group. DS: degree of substitution; DTT: 1,4-dithiotreitol; gel-MA: methacrylated gelatin; gel-NB: norbornene-functionalized gelatin; gel-SH: thiolated gelatin; LAP: lithium(2,4,6-trimethylbenzoyl) phenylphosphinate.

Hydrogel	Concentration	DS	Photo-Initiator	Cross-Linker	Abbreviation
Gel-MA	10% *w*/*v*	97%	2 mol% LAP	NA	Gel-MA
Gel-NB	10% *w*/*v*	85%	2 mol% LAP	DTT	Gel-NB85/DTT
Gel-NB	10% *w*/*v*	55%	2 mol% LAP	DTT	Gel-NB55/DTT
Gel-NB	10% *w*/*v*	85%	2 mol% LAP	Gel-SH 75%	Gel-NB85/SH75

## Data Availability

The data presented in this study are available on request from the corresponding author.

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
