# Peer review of "Equine Tenocyte Seeding on Gelatin Hydrogels Improves Elongated Morphology"

_polymers, 2021, doi:10.3390/polym13050747_

Round 1

Reviewer 1 Report

This manuscript describes the characterization of gelatin-based hydrogels  fabricated with different crosslinking strategies. The physico-chemical properties are analyzed in relation to tenocyte viability and morphology. The methodology described here is adequate and seems to be simple and appropriate for general use in cellular studies and bioengineering.  Overall, manuscript data appears to be correct and thorough.

Author Response

We would like to thank the reviewers for their thorough reading and constructive remarks. In response to their suggestions, we have modified the manuscript to make it a more concise story. Since we had the opportunity to clarify some issues, we believe the quality of the manuscript has substantially improved. Therefore, it is our hope that this revised version of our manuscript will be acceptable for publication in Polymers’ special Issue “Bio-Inspired (Nano)Structured Polymer Scaffolds, Bio-Adhesives, and Surfaces.

Reviewer 2 Report

In this study, the physicochemical properties, and cellular functionality of methacrylated gelatin, norbornene functionalized gelatin, and thiolated gelatin were compared. The study included the characterization of degree of substitution, gel fraction, swelling ratio, storage modulus, cell proliferation, viability and morphology. The topic is interesting and relevant for the scientific community. Overall the manuscript is well structured and can be considered to publish after some minor revisions.

  1. In section 2. Materials and methods, please include the information of the supplier’s name and location of gelatin, methacrylic anhydride, EDC, NHS, DMSO, 5-norbornene-2-carboxylic acid, acetone, EDTA, LAP, etc.
  2. Figure 1 indicate that the starting material of Gel-MA is gelatin type B, whereas the description of the method to develop gel-MA (section 2.1.1) only specify that the starting material is gelatin. Please clarify.
  3. Please add the NMR spectra graphs to support the DS results of Gel-MA, and Gel-NB.
  4. Figure 3 shows the physico-chemical properties (gel fraction, mass swelling ratio, and storage modulus) of the obtained hydrogel films. The charts are utilized to compare the performance of the Gel MA, GELNB85DTT, GELNB55DTT, and GELNB85SH75 hydrogels. The upper limit error bars of the presented in the charts but the lower limit error bars are not shown. Please add the lower limit error bars so that the comparison of the gel properties can be clearly evidenced.

Reviewer 3 Report

The manuscript EQUINE TENOCYTE SEEDING ON GELATIN HYDROGELS IMPROVES ELONGATED MORPHOLOGY describes a comparison of GELMA and GELNB cross-linked with 1,4-dithiotreitol (DTT) or thiolated gelatin (gel-SH) for the potential treatment of tendinopathy. The manuscript does not present the aim of the study clearly and the amount of results is a bit low. However, the study could be improved by including some changes.

Results:

Figure 4 is very confusing, the font must be bigger and it is very difficult to read and understand. I suggest to make two different figures to explain the results better. 

Figure 5 is not very clear, images at higher magnifications must be taken to study the morphology of the cells. In addition, the number of replicates and the details in the measurements must be included to confirm the morphological changes of the cells. However, qPCR studies must be perform to confirm the evolution to tenocytes, if not possible at least immunostaining of collagen type IV and I.

Discussion: as mentioned before some of the results must be improved and therefore the discussion should be changed in the manuscript.

Conclusions: please create a specific section for that as the final conclusion is not clear.
